# Map Connectivity and Empirical Hardness of Grid-based Multi-Agent Pathfinding Problem

**Primary Keywords:** *Multi-Agent Planning*

## Abstract

We present an empirical study of the relationship between map connectivity and the empirical hardness of the multi-agent pathfinding (MAPF) problem. By analyzing the second smallest eigenvalue (commonly known as $\lambda_2$) of the normalized Laplacian matrix of different maps, our initial study indicates that maps with smaller $\lambda_2$ tend to create more challenging instances when agents are generated uniformly randomly. Additionally, we introduce a map generator based on Quality Diversity (QD) that is capable of producing maps with specified $\lambda_2$ ranges, offering a possible way for generating challenging MAPF instances. Despite the absence of a strict monotonic correlation with $\lambda_2$ and the empirical hardness of MAPF, this study serves as a valuable initial investigation for gaining a deeper understanding of what makes a MAPF instance hard to solve.

## Introduction

Multi-agent pathfinding (MAPF) is the problem of finding collision-free paths for a team of agents on a map from a set of start positions to a set of goal positions (Stern et al. 2019a). Given an undirected map, an optimal MAPF algorithm computes the minimum path cost for all the agents such that no two agents occupy the same location or traverse the same edge at an identical time step. Although solving MAPF optimally is proven to be NP-Hard (Yu and LaValle 2013), many real-world MAPF instances can be solved optimally within a reasonable time. While optimal MAPF algorithms can solve some instances with hundreds of agents, they can also struggle on instances with only a small number of agents (Ren et al. 2021; Ewing et al. 2022).

We are interested in understanding what features of MAPF instances make them hard to be solved optimally. We are also interested in finding an effective way to compare the hardness of different maps when randomly generating MAPF instances on them. For example, when using uniform random sampling to generate agents and goals on two given maps, we seek to predict which map will have harder instances on average. This area of research is known as *empirical hardness*, which focuses on identifying features that determine how hard individual instances will be for particular algorithms to solve (Leyton-Brown, Nudelman, and Shoham 2009). Here, we present an empirical study that aims to elucidate the correlation between map connectivity and the em-

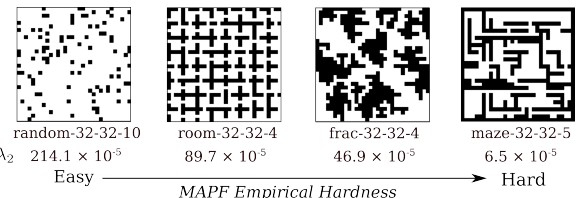

$$\lambda_2 \quad \begin{array}{cccc} \text{random-32-32-10} & \text{room-32-32-4} & \text{frac-32-32-4} & \text{maze-32-32-5} \\ 214.1 \times 10^{-5} & 89.7 \times 10^{-5} & 46.9 \times 10^{-5} & 6.5 \times 10^{-5} \end{array}$$

Easy ⟶ Hard
*MAPF Empirical Hardness*

Figure 1: Example maps and their $\lambda_2$

pirical hardness of the multi-agent pathfinding problem.

There are two major components of a MAPF instance: the map topology and distribution of the agents. In this study, we focus on 2D grid-based MAPF problems, where a map can be viewed as a 4-connected graph $G(V, E)$. In spectral graph theory, the second smallest eigenvalue of the normalized Laplacian matrix (henceforth referred to as $\lambda_2$) of $G(V, E)$ serves as an algebraic measurement of graph connectivity. Figure 1 shows the value of $\lambda_2$ for several maps with different connectivity. The difference in $\lambda_2$ between a well-connected map, `random-32-32-10` on the far left and a less connected `maze-32-32-5` on the far right is significant.

In this paper, we present empirical results that show the $\lambda_2$ of $G(V, E)$ is correlated with the empirical hardness of MAPF instances generated using uniform random sampling for agents and goals. While a smaller $\lambda_2$ value does not consistently yield challenging instances, instances characterized as difficult tend to occur more frequently when $\lambda_2$ is small. The most straightforward small $\lambda_2$ example is a map with many narrow corridors. Previous research has shown that various optimal MAPF algorithms have difficulty with such maps even with a small number of agents (Li et al. 2020; Ren et al. 2021), which could be caused by the over-congestion and conflicts that narrow corridors bring.

We also propose a map generator based on Quality Diversity (QD) (Mouret and Clune 2015) which provides the flexibility to generate maps within a desired range of $\lambda_2$. This provides an effective way to find maps that might be challenging for MAPF algorithms or generating benchmark dataset that covers a greater spectrum of connectivity.

Although $\lambda_2$ does not exhibit a strict monotonic correlation with empirical hardness, this study serves as a valuable initial study for understanding MAPF empirical hardness.

## Preliminary

### Normalized Laplacian and Cheeger's Inequality

In spectral graph theory, the normalized Laplacian matrix $\bar{L}$ of a graph is defined by:

$$L = D - A$$
$$\bar{L} = D^{-1/2}LD^{-1/2} = I - D^{-1/2}AD^{-1/2} \quad (1)$$

where the $D$ is the diagonal degree matrix and $A$ is the adjacency matrix. The second smallest eigenvalue of the normalized Laplacian $\bar{L}$ defines the algebraic connectivity of the graph, describing how well the graph is connected.

To get a better understanding of why $\lambda_2$ is related to the connectivity of graphs, we first introduce the *boundary* for a set of vertices $S \subset V$ of undirected graph $G(V,E)$:

$$\partial S = \{\{i,j\} \in E : i \in S, j \notin S\}. \quad (2)$$

The *conductance* of $S$ is defined as:

$$\phi(S) = \frac{|\partial S|}{\min(d(S), d(V\backslash S))} \quad (3)$$

where $|\partial S|$ is the number of edges on the boundary and $d(S)$ denotes the number of edges with both endpoints (nodes) within $S$. The $\phi(S)$ represents the ratio of the number of edges on the boundary of set $S$ to the minimum of its internal and external edges.

The *conductance* of a graph $G(V,E)$ is subsequently defined as the smallest conductance over all cuts, where cuts refer to partitions of vertices:

$$\phi(G) = \min_{\emptyset \subsetneq S \subsetneq V} \phi(S) \quad (4)$$

The conductance represents how well-connected a graph is.

**Theorem 1.** (Cheeger's Inequality). Let $\lambda_2$ be the second smallest eigenvalue of the normalized Laplacian $\bar{L}$ of undirected graph $G(V,E)$, then:

$$\frac{\lambda_2}{2} \leq \phi(G) \leq \sqrt{2\lambda_2}. \quad (5)$$

Cheeger's inequality brings the graph connectivity and $\lambda_2$ together. This implies that $\lambda_2$ can be used as a quantitative method for characterizing the impacts of a map's features, such as narrow corridors, on the overall map connectivity. Generally, a relatively small $\lambda_2$ indicates the graph is poorly connected, whereas a large $\lambda_2$ implies strong connectivity (for more detail please refer to (Vidick 2018)).

### Conductance and MAPF Conflicts

Here we present an intuitive proof of how the maps with smaller conductance are more likely to generate more conflicts for MAPF instances. Consider a simple dumbbell graph $G_d$ shown in Figure 2(a), where two partitions are only connected with a single edge. The size of the circle indicates different number of edges within the partition. Let's also assume this partition $S^*$ has the smallest conductance of $G_d$, thus we have $\phi(G_d) = \phi(S^*)$. Next, consider another partition $S$ of $G_d$ shown in Fig. 2(b), where the two partitions are connected by more edges; thus, $|\partial(S)| > 1$

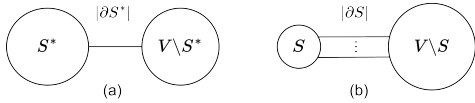

Figure 2: A dumbbell graph with two different partitions.

and $\phi(S^*) < \phi(S)$. Another observation is that $S^*$ is a more balanced partition than $S$ in terms of the number of edges within the partition, and we further have:

$$d(S^*)d(V\backslash S^*) > d(S)d(V\backslash S). \quad (6)$$

When uniformly and randomly sampling the start and goal locations on $G_d$, the shortest path will traverse a boundary edge only if the start and goal locations are on different sides of the boundary. The probability of the shortest path visiting a boundary edge of partition $S$ is:

$$P(\partial S) = \frac{1}{|\partial S|}\frac{2d(S)d(V\backslash S)}{d(V)^2} \quad (7)$$

Given Eq. 6, we have:

$$\frac{2d(S^*)d(V\backslash S^*)}{d(V)^2} > \frac{2d(S)d(V\backslash S)}{d(V)^2} > \frac{1}{|\partial S|}\frac{2d(S)d(V\backslash S)}{d(V)^2}.$$

The left-hand side is $P(\partial S^*)$ since $|\partial S^*| = 1$ and the right-hand side is $P(\partial S)$. This indicates $P(\partial S^*) > P(\partial S)$. This inequality implies a higher likelihood of agents visiting the boundary edges of poorly connected cuts within the same graph, leading to increased potential conflicts, particularly in scenarios with more agents. Intuitively, one can think of these boundary edges as choke points that need to be traversed to get from one partition to the other. Relating this to the definition of $\phi(G)$ and Cheeger's inequality, we can loosely demonstrate that $P(\partial S^*) \propto \frac{1}{\phi(G)}$. This suggests that maps with smaller $\phi(G)$ or $\lambda_2$ may tend to exhibit more conflicts; thus MAPF instances on those maps are more likely to be challenging.

## Quality Diversity Instance Generator

To investigate the relationship between $\lambda_2$ and the empirical hardness of maps, we developed a map generator that can produce maps with a given $\lambda_2$ value. The maps generated should provide as much diversity as possible in measures other than $\lambda_2$ to try and isolate the relationship between $\lambda_2$ and hardness. We used a Quality Diversity (QD) method based on the algorithm MAP-Elites (Mouret and Clune 2015). MAP-Elites is a search space illumination algorithm that seeks to find high quality solutions that are diverse along certain prescribed features. It maintains a container, which contains a set of potential solutions. New potential solutions are inserted into the container if they are the best solution found so far in a specific bin of feature space with respect to some objective function. MAP-Elites typically utilizes evolutionary algorithms to alter existing solutions in the container and produce new potential solutions.

For our purposes, we sought to produce maps with specific $\lambda_2$ values that had diverse obstacle properties. We set

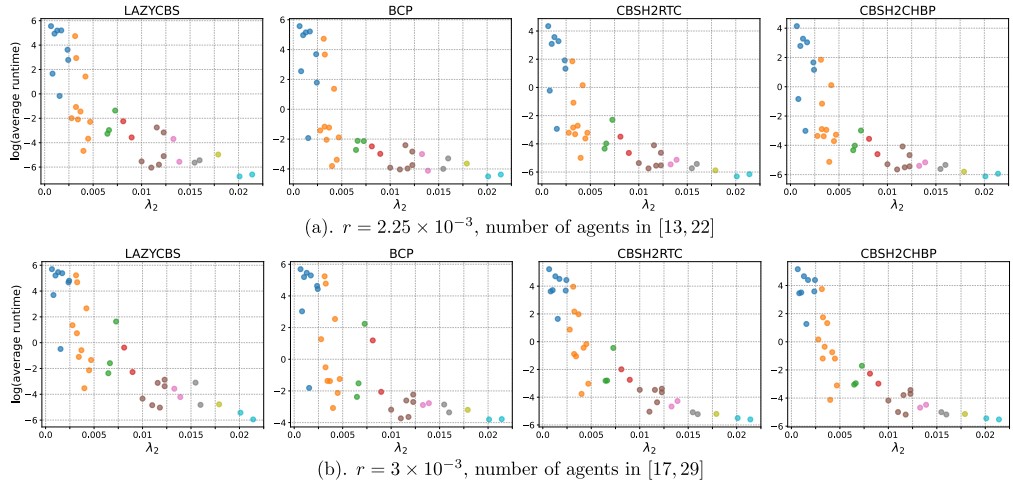

(a). $r = 2.25 \times 10^{-3}$, number of agents in $[13, 22]$

(b). $r = 3 \times 10^{-3}$, number of agents in $[17, 29]$

Figure 3: Simulation results for the logarithm of average runtime and $\lambda_2$ for various maps, with distinct color coding denoting different ranges of $\lambda_2$.

the objective function to be distance from a desired $\lambda_2$ value. We used features on the percentage of obstacles and the density of those obstacles (how many obstacles were adjacent to other obstacles). For each iteration, we took an existing map from the container and with equal probability we either "mutated" the map or "crossed" the map with another random map from the container. A mutation consisted of adding or removing up to five obstacles on the map uniformly at random. Crossing two maps involved randomly selecting regions of one map to add to the other map. We then checked for connectedness and added back the minimum number of vertices to reconnect any disconnected components. The new maps, either with randomly added or removed obstacles or the cross between existing maps, were then evaluated on closeness to the desired $\lambda_2$ value. If they were closer than any other map with similar features, they were kept and the other map in the container with those feature values was removed.

Previous work has explored generating MAPF maps with Quality Diversity algorithms to generate maps suitable for high-throughput online MAPF (Zhang et al. 2023). Zhang et al. trained a surrogate model DSAGE (Bhatt et al. 2022) that could help repair instances to meet constraints (e.g., number of shelves and connectivity) and predict the throughput of an instance. Our problem requires less sophisticated repair, since we have no hard constraint on the number of obstacles in an instance. Additionally, our objective function is relatively easy to compute, requiring only $\lambda_2$ for a generated map, and does not require any additional MAPF simulations. Our map generator is designed to create instances with a wide range of connectivity to use in the evaluation and benchmarking of MAPF algorithms.

## Experiments

To thoroughly investigate the relationship between map connectivity and empirical hardness of MAPF, we have selected four different optimal MAPF algorithms which are

proven to be quite powerful according to various benchmark analysis (Ewing et al. 2022; Shen et al. 2023a): **LazyCBS** (Gange, Harabor, and Stuckey 2019), **BCP** (Lam et al. 2022), **CBSH2-RTC** (Li et al. 2021) and **CBSH2-RTC-CHBP** (Shen et al. 2023b).

**Simulation Setup.** To ensure the diversity of our test dataset, we included maps from multiple data sources. Firstly, we have included all $32 \times 32$ maps (5 in total) from the MAPF benchmark dataset (Stern et al. 2019b). Additionally, apart from our QD map generator, we have also included a fractal map generator based on diffusion-limited aggregation method (Ewing et al. 2022). We slightly modified the generation rule of the fractal method such that it could generate maps of different styles (e.g., cave-like `frac-32-32-4` and maze-like `maze-32-32-5` in Figure 1). We generated 31 fully-connected maps of size $32 \times 32$ using QD and fractal generator. Please refer to the Supplementary Material for more detailed map information.

When generating MAPF instances, we ensured that all the maps have the same agent-to-freespace ratio, where $r = \frac{\#agents}{\#free\ cells}$. This value is chosen based on our test such that the instances are neither excessively challenging nor overly easy so that we can still effectively compare the performance across different maps. For each map, we generated 100 instances using uniform random sampling to determine the start and goal locations of agents. The feasibility of the generated instances was validated by using a sub-optimal MAPF algorithm ECBS with a relaxed bound ($w = 1.6$) (Barer et al. 2014). Simulations were conducted on a PC with Ryzen 3950x CPU and 64GB RAM, with the runtime limit set to 300 seconds.

**Experiment 1: Average Runtime and $\lambda_2$.** As an initial proof of concept to show that the $\lambda_2$ value of a map has some correlation with the empirical hardness, or runtime, of MAPF instances on that map, we randomly generated MAPF instances on 36 maps with varying $\lambda_2$ values and

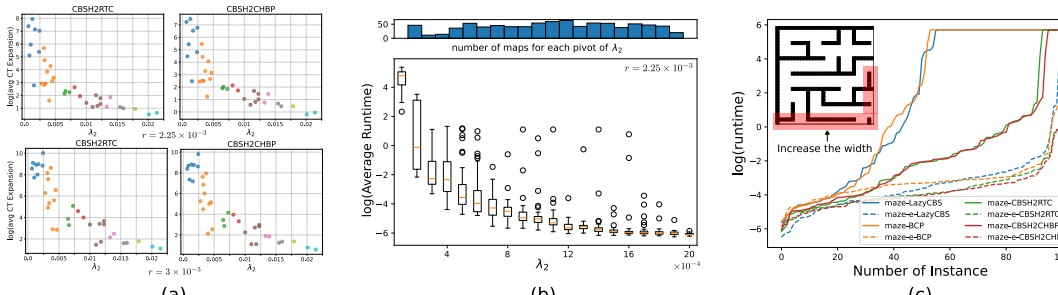

Figure 4: (a). The logarithm of average number of CT expansions and $\lambda_2$ and for different maps. (b). Boxplot for $\lambda_2$ and the logarithm of average runtime for the maps created by QD generator. (c). Sorted logarithm of runtime for `maze` and its expanded version `maze-e` by increasing the width of the narrow corridors in red boxes from 1-cell to 2-cell.

compared runtimes. The simulation results in Fig. 3 illustrate the relationship between the logarithm of average runtime and $\lambda_2$ of different maps. We have made several interesting observations on the results.

First, hard instances often appear on maps with smaller $\lambda_2$ (top left corner), whereas maps with larger $\lambda_2$ can be considerably easy (bottom right corner). This pattern remains consistent across different algorithms and $r$ settings. Additionally, it is noteworthy that CBSH2-based algorithms generally exhibit faster runtime than LazyCBS and BCP (notice the different scales on y-axis). Despite differences in absolute runtime, our results indicate within each algorithm, challenging instances happen more frequently on maps with smaller $\lambda_2$.

Second, maps with smaller $\lambda_2$ could still have relatively easy instances. Given that $\lambda_2$ is not the only factor influencing empirical hardness, we are not surprised to see that the average runtime and $\lambda_2$ do not exhibit a strict monotonic correlation. One possible reason might be the effect of narrow corridors on a 2D grid-map, for instance increasing the width of a narrow corridor from 1-cell to 2-cell width will only change $\lambda_2$ slightly, but the wider corridors are less likely to create enough contested regions, thus the empirical hardness could drastically shift from hard to easy. We further explore this in Experiment 4.

**Experiment 2: Average Number of Constraint Tree (CT) Expansions and $\lambda_2$.** Next, we illustrate the relationship of the average number of CT expansions and $\lambda_2$ for all instances on a map. For the CBSH2-based algorithms, the number of CT expansions is related to how many conflicts have been resolved during the searching process and reflects the hardness of instances. From Figure 4(a), the trend for number of CT expansions is similar to the runtime trend. This correlation is believed to be caused by the poorly connected regions where conflicts are more likely to happen.

**Experiment 3: More Tests Using QD Map Generator.** Here we present additional tests on the runtime using CBSH2-RTC for maps generated by our QD map generator. We have generated $851$ maps with a step size of $10^{-4}$ for $\lambda_2$ and the number of maps for each pivot is shown in Fig. 4(b). Different from fractal map generator, which lacks control over map connectivity, the QD map generator is able to generate maps with a well-distributed range of $\lambda_2$. This nice feature makes it a great choice to create benchmark dataset that requires a wider spectrum of map connectivity. Despite there are still many outliers in Figure 4(b), the relationship between $\lambda_2$ and empirical hardness still holds. Indicates that hard instances tend to happen around small $\lambda_2$, while large $\lambda_2$ generally result in easier instances.

**Experiment 4: Expand the Width of Narrow Corridors.** In Experiment 1, we mentioned that there are still many easy instances on maps with low $\lambda_2$. To investigate this phenomenon, we manually changed the connectivity of maps without affecting the number of obstacles. More specifically, we expand some of the narrow corridors (red boxes in Figure 4(c)) in a `maze` map from 1-cell to 2-cell width and observed that $\lambda_2$ changed from $10.1 \times 10^{-5}$ to $15.4 \times 10^{-5}$. Although the change in $\lambda_2$ is small, there is a significant change in empirical hardness, where instances on expanded version `maze-e` (shown with dashed lines) are much easier. This demonstrates that even maps with small $\lambda_2$ can have easy instances. It also suggests that a 1-cell-width corridor is more likely to create contested regions and cause conflicts between agents, thus slowing down the algorithms (especially for conflict-based algorithms). These contested regions are significantly mitigated when increasing the corridor width, making the instances easier; in the meantime $\lambda_2$ exhibits minor change. We intend to develop a hybrid reasoning on both $\lambda_2$ and corridor width in future research.

## Conclusion

In summary, even though $\lambda_2$ does not exhibit a strict monotonic correlation with empirical hardness, it still shows notable effectiveness, especially for very challenging instances associated with small $\lambda_2$. Considering the simplicity and ease of comparing $\lambda_2$ across different maps, we believe it is a reasonably effective metric and great starting point for future research on MAPF empirical hardness. Another contribution of this work is the QD map generator which can generate maps with the desired range of $\lambda_2$. Possible future works are developing more powerful MAPF instance generators with tunable empirical hardness and providing a more precise theoretical bound on the correlation of $\lambda_2$ and empirical hardness of MAPF problem.

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
