# OpenReview forum: "Map Connectivity and Empirical Hardness of Grid-based Multi-Agent Pathfinding Problem"
_icaps-conference.org/ICAPS/2024/Conference — ICAPS 2024_

### Official Review · Reviewer_ghz5 · 2023-12-25

**Significance And Importance:** 2
**Soundness:** 4
**Novelty:** 2
**Clarity:** 4
**Overall Evaluation:** 2
**Confidence:** 4

**Weaknesses:**

2: No major or minor weaknesses.

**Contributions Of The Paper:**

This short paper presents an empirical study regarding the complexity of MAPF problems. Such problems have attracted significant interest in the past decade and results are split between presenting general hardness results for variants of the problem and suggesting algorithms that work well in practice. This paper is the first to suggest any empirical hardness results by connecting the second smallest eigenvalue of the normalized Laplacian matrix of different maps with the hardness depicted by existing algorithms.

**Ethical Considerations:**

(1) Not Applicable: The paper does not have any ethical considerations to address

**Nomination For Best Paper:**

No

**Questions For Authors:**

Will the authors release all of their code?

**Reproducibility:**

2: Some details are missing, but the paper still appears to be replicable with some effort.

**Strengths Of The Paper:**

While the results are not conclusive, this is to be expected as there are likely more than one factor affecting the empirical running time. However, as the authors showcase, their work does present an interesting non-trivial correlation. I believe that the community can benefit from these types of results and would like to see this paper published. It is written well and I have no major reservations regarding the work.

**Weaknesses Of The Paper:**

Small comments:
(*) Punctuate figure captions as well as equations.
(*) Line 91 - remove "The"
(*) Line 276- "Despite \insert{the fact that}...
(*) Line 278- \insert{This} indicates...
(*) Gordon et al [1] discuss the complexity of CBS' CT. This should be mentioned in the context of experiment 2 (I found no immediate insight from their work relevant to this paper but it is still relevant nonetheless).

[1] Ofir Gordon, Yuval Filmus, Oren Salzman: Revisiting the Complexity Analysis of Conflict-Based Search: New Computational Techniques and Improved Bounds. SOCS 2021: 64-72

---

> ### Author Rebuttal · Authors · 2024-01-28
>
> Q1. Will the authors release all of their code?
>
> A1. Yes, all code will be released open source as soon as possible.
>
> We will also make sure to address all of the small comments from the reviewer, including a discussion of Gordon et al.[1] on the complexity of CBS' Constraint Tree to provide more background information on Experiment 2.
>
> [1] Ofir Gordon, Yuval Filmus, Oren Salzman: Revisiting the Complexity Analysis of Conflict-Based Search: New Computational Techniques and Improved Bounds. SOCS 2021: 64-72
>
> Again, we thank all of the reviewers for their insightful comments and for emphasizing the potential impacts of our work to the MAPF community.

---

### Official Review · Reviewer_nHnS · 2024-01-17

**Significance And Importance:** 1
**Soundness:** 2
**Novelty:** 2
**Clarity:** 2
**Confidence:** 5

**Weaknesses:**

0: Minor weaknesses requiring some work to be addressed for the paper to be accepted.

**Contributions Of The Paper:**

This paper proposes to use the second smallest eigenvalue of the normalized Laplacian matrix of MAPF instances, to predict the empirical hardness of an instance for four exact MAPF solvers and use it for generating difficulty-controlled MAPF instances. However, the authors clarify that this feature does not show a strong correlation with the runtimes of all solvers and may be used together with other features that are not clearly studied. Overall, the experiments do not allow for drawing strong conclusions.

**Ethical Considerations:**

(5) Excellent: The paper comprehensively addresses all of the applicable ethical considerations

**Nomination For Best Paper:**

No

**Overall Evaluation:**

-1: (weak reject)

**Questions For Authors:**

Why didn't you try to compare the feature you propose to other, simpler, metrics, already documented in the literature?

**Reproducibility:**

3: Authors describe the implementation and domains in sufficient detail.

**Strengths Of The Paper:**

- The paper deals with an important problem for MAPF solving... Predicting the hardness of an instance.
- The paper proposes an unstudied feature within this context, showing promising results.

**Weaknesses Of The Paper:**

- The results do not allow to draw strong conclusions.
- I think some simple baselines have to be compared with the proposed feature in order to measure its general utility. For example, previous research has already shown that the overall density of an instance (#obstacles + #agents) / #cells, together with the number of connected components in the map, can be used as a very simple feature to predict its hardness.

---

> ### Author Rebuttal · Authors · 2024-01-28
>
> Q1. Why didn't you try to compare the feature you propose to other, simpler, metrics, already documented in the literature?
>
> A1. This is a great question! Our study on  $\lambda_2$ actually stems from the fact that there are no effective simple metrics for comparing empirical hardness without considering map connectivity. Given the same number of agents, obstacles, and free map cells, the empirical hardness could still be significantly different based on where the obstacles are located on a map.
>
> We are aware of the metric suggested in the review, namely "(# obstacles + #agents) / #cell, together with the number of connected components in the map". However, our research indicates that it is not a very effective estimate of the empirical hardness of the grid-based MAPF problem. The main concern regarding this metric is that it does not reflect enough features or information about map connectivity. In our current experiment setting, all maps are fully connected, so the number of connected components in the map is always 1.
> As for "(# obstacles + #agents) / #cells'', many simple counter-examples can be created that show that this metric fails to consistently predict empirical hardness.
> For instance, consider two maps with the same value of "(# obstacles + #agents) / #cells".
> The first map, with all obstacles on the border of the map, would be very "easy", as the map is mostly empty and highly connected.
> On the other hand, another example can be formulated by placing obstacles vertically in the middle of the map, like a wall, with only a 1-cell wide gap (that is, a horizontal corridor) in it. Such a map would result in much harder instances. Fig.3 of the supplementary material also shows several hard maps with bad connectivity following similar logic. In the Preliminary Study section, we also briefly discussed why a poorly connected map like this will likely lead to much harder MAPF instances.
>
> Due to the page limit, we could not include this information in the main text, but we will add a few more examples in the supplementary material where maps have exactly the same number of obstacles, agents, and free cells, but the empirical hardnesses are distinctively different. We thank the reviewer for pointing this out.

---

### Official Review · Reviewer_rWSq · 2024-01-22

**Significance And Importance:** 2
**Soundness:** 3
**Novelty:** 2
**Clarity:** 3
**Confidence:** 4

**Weaknesses:**

1: Minor weaknesses that are easily fixable.

**Contributions Of The Paper:**

This paper suggests using \lambda2 (which is known to provide a measure of the connectivity of a graph) as a measure of the hardness of a MAPF instance. To better assess on their conjecture the authors devised a system that can create instances with almost any \lambda2 value.

**Ethical Considerations:**

(1) Not Applicable: The paper does not have any ethical considerations to address

**Nomination For Best Paper:**

No

**Overall Evaluation:**

-1: (weak reject)

**Questions For Authors:**

Unless I'm wrong, \lambda2 is restricted to undirected graphs. Is this true? i.e., do your conjecture apply only to undirected graphs?

What is the number of agents tried in the experimental section? I found no information about this anywhere

**Reproducibility:**

3: Authors describe the implementation and domains in sufficient detail.

**Strengths Of The Paper:**

Putting everything together: \lambda2 measures the connectivity in a graph, and the authors suggest to use \lambda2 to assess on the difficulty of solving specific MAPF instances, i.e., they conejcture that connectivity is the major factor playing a role in the difficulty of a MAPF instance. Indeed, their results somehow support their claims and as they themselves say, even if the correlation is not perfect it is undeniable that there is some correlation there that might be worth researching deeper.

**Weaknesses Of The Paper:**

I think that connectivity is an easy-to-catch factor affecting difficulty of a MAPF instance, but I'm rather surprised that the way \lambda2 is measured does not take the location of agents into account. The example most repeated is that of corridors where obviously agents have trouble to walk through. On the other hand, making the corridors wider obviously help any agent to solve an instance more easily, but I think that there are some concepts which are not correctly captured by the proposed measure. Well, as the authors themselves say the correlation is not perfect so that wondering what might be the other factors is also relevant. I understand that difficulty is not relative to the number of agents, because as it increases problems become harder, the point is that some might become harder than others. Agents play a role definitely, not the number, but the paths they have to follow, and no information about the number of agents is provided in the experiments (other than the agent-to-freespace ratio); on the other hand,the number of CTs clearly plays a role in the difficulty of an instance (the advantage of \lambda2 being that it can be computed beforehand!) but this is clearly affected by the number of agents.

Sincerely, I'm a little bit surprised that hardness progresses linearly with a measure such as \lambda2 and I would have expected instead to see some kind of transition phase somewhere.

---

> ### Author Rebuttal · Authors · 2024-01-28
>
> A1. In the context of this paper, our $\lambda_2$ analysis is restricted to the undirected graph. Since the map of the vanilla grid-based MAPF problem can always be viewed as an undirected graph, this setting is sufficient. However, Cheeger's inequality can also be applied to directed graphs[1], which makes it possible to apply $\lambda_2$ analysis to directed graphs and more complicated variants of MAPF problems. [1] Chung, F. Laplacians and the Cheeger Inequality for Directed Graphs. 2005.
>
> A2. In Experiment 1, the number of agents ranges from 13 to 22 for $r=2.25 \times 10^{-2}$, and 17 to 29 for $r=3 \times 10^{-2}$ (a typo in Fig.3 sub-caption on $r$ will be corrected). For Experiment 3, with $r=2.25 \times 10^{-2}$, it ranges from 19 to 25. We will make sure to clarify this in the main text.
>
> We would also like to provide a more comprehensive explanation of the reviewer's comment that "the way $\lambda_2$ is measured does not take the location of agents into account".
>
> This is the case mainly because $\lambda_2$ is only related to the map connectivity or topology rather than the agent locations. The reviewer's comment about this map feature not being the only factor on empirical hardness is entirely correct. As we mentioned in the paper, map topology and agent locations are both major components of a MAPF instance, and both affect the empirical hardness of an instance. However, it is non-trivial to consider these two factors together without studying them individually first.
>
> We chose to study the map feature first since uniformly randomly distributed agents are used in many benchmark instances, and we are interested in general performance on different maps. Given a set of maps, how can we tell which map is harder if many MAPF instances will be run on it? Being able to effectively approximate map hardness is very useful in many applications. For example, when designing a layout for a warehouse where many MAPF instances will be executed, how can one effectively determine which layout is better? Or, when designing a MAPF benchmark dataset, how can one ensure that it contains maps of different hardness? Additionally, if users decide to generate agents' locations using other means, how to guarantee a sufficient number of challenging instances for these maps?
> Our study provides a proof-of-concept for analyzing empirical hardness and aims to inspire future work. We're very thankful for the reviewer's comment that enriched our discussion on this work.

---

### Meta-Review · Area_Chair_NAqZ · 2024-02-06

**Recommendation:** Accept (Oral)
**Confidence:** 4

**Metareview:**

This (short) paper attempts to address a very important and non-trivial question -- what makes a multi-agent pathfinding (MAPF) instance empirically hard to solve (using state-of-the-art solvers). Intuitively, there are several primary factors affecting the empirical hardness including map topology, the density of agents etc. The authors focus on one such factor, i.e. on the map connectivity, and demonstrate how the latter correlates with the empirical outcomes.

The reviewers are in agreement that the number of agents (density of agents) is, clearly, the factor that pays an important role in establishing MAPF hardness and disregarding this factor (as the authors did) requires justification. On the other hand, analyzing how map connectivity affects hardness of a MAPF instance when the density of agents is fixed, may still provide value to the community and may spark a further conversation.

**Ethical Considerations:**

(1) Not Applicable: The paper does not have any ethical considerations to address